# Chirality Nets for Human Pose Regression

**Raymond A. Yeh,**[*] **Yuan-Ting Hu**[*], **Alexander G. Schwing**
Department of Electrical Engineering, University of Illinois at Urbana-Champaign
{yeh17, ythu2, aschwing}@illinois.edu

## Abstract

We propose Chirality Nets, a family of deep nets that is equivariant to the "chirality transform," *i.e.*, the transformation to create a chiral pair. Through parameter sharing, odd and even symmetry, we propose and prove variants of standard building blocks of deep nets that satisfy the equivariance property, including fully connected layers, convolutional layers, batch-normalization, and LSTM/GRU cells. The proposed layers lead to a more data efficient representation and a reduction in computation by exploiting symmetry. We evaluate chirality nets on the task of human pose regression, which naturally exploits the left/right mirroring of the human body. We study three pose regression tasks: 3D pose estimation from video, 2D pose forecasting, and skeleton based activity recognition. Our approach achieves/matches state-of-the-art results, with more significant gains on small datasets and limited-data settings.

## 1 Introduction

Human pose regression tasks such as human pose estimation, human pose forecasting and skeleton based action recognition, have numerous applications in video understanding, security and human-computer interaction. For instance, collaborative virtual reality applications rely on accurate pose estimation for which significant advances have been reported in recent years.

Specifically, recent state-of-the-art approaches use supervised learning to address pose regression and employ deep nets. Input and output of those nets depend on the task: inputs are typically 2D or 3D human pose key-points stacked into a vector; the output may represent human pose key-points for pose estimation or a classification probability for activity recognition. To improve accuracy of those tasks, a variety of deep net architectures have been proposed [34, 3, 17, 29, 42, 48], generally relying on common deep net building blocks, such as, fully connected, convolutional or recurrent layers. Unlike for image datasets, to enlarge the size of human pose datasets, a reflection (left-right flipping) of the pose coordinates as illustrated in step (1) of Fig. 1 is not sufficient. The chirality of the human pose requires to additionally switch the labeling of left and right as illustrated in step (2) of Fig. 1.

However, while this two-step data augmentation is conceptually easy to employ during training, we argue that even better accuracy is possible for human pose regression tasks if this pose symmetry is directly built into the deep net. In particular, if confronted with either of the poses illustrated on the left or right hand side of Fig. 1 the output of a deep net should be equivariant to the transformation, *i.e.*, the output is also transformed in a "predefined way." For example, if the network's output is also a human pose, the output pose should follow the same transformation. On the other hand, for an activity recognition task, the output probability should remain unchanged. The equivariant map, for pose estimation, is illustrated in Fig. 2 and we make the equivariance property more precise later.

To encode this form of equivariance for human pose regression tasks, we propose "chirality nets." Specifically, the output of a chirality net is guaranteed to be equivariant w.r.t. a transformation composed of reflections and label switching. To build chirality nets, we develop chirality equivariant

---

[*]Indicates equal contribution.

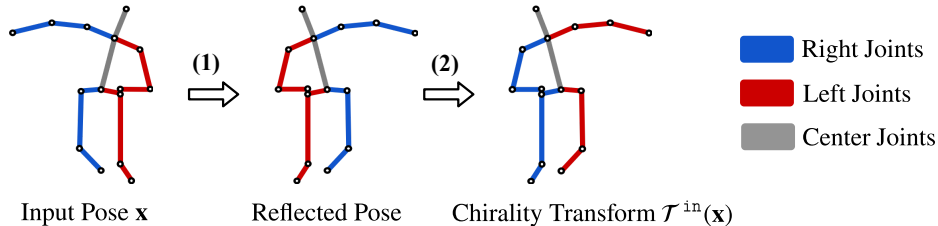

Figure 1: Illustration of the chirality transformation. The transformation includes two operations, (1) a reflection of the pose, *i.e.*, a negation of the x-coordinates; and (2) a switch of the left / right joint labeling. The ordering of the two operations are interchangeable.

versions of commonly used layers. Specifically, we design and prove equivariance for versions of fully connected, convolutional, batch-normalization, dropout, and LSTM/GRU layers and element-wise non-linearities such as tanh or soft-sign.The main common design principle for chirality equivariant layers is odd and even symmetric sharing of model parameters. Hence, in addition to being equivariant, transforming a typical deep net into its chiral counterpart results in a reduction of the number of trainable parameters, and lower computation complexity due to the symmetry in the model weights. We find a smaller number of trainable parameters reduces the sample complexity, *i.e.*, the models need less training data.

We demonstrate the generalization and effectiveness of our approach on three pose regression tasks over four datasets: 3D pose estimation on the Human3.6m [22] and HumanEva dataset [49], 2D pose estimation on the Penn Action dataset [64] and skeleton-based action recognition on Kinetics-400 dataset [23]. Our approach achieves state-of-the-art results with guarantees on equivariance, lower number of parameters, and robustness in low-resource settings.

## 2    Related Work

First we briefly review invariance and equivariance in machine learning and computer vision as well as human pose regression tasks.

**Invariant and equivariant representation.** Hand-crafted invariant and equivariant representations have been utilized widely in computer vision systems for decades, *e.g.*, scale invariance of SIFT [32], orientation invariance of HOG [9], affine invariance of the Harris detector [36], shift-invariant systems in image processing [54], *etc*.

These properties have also been adapted to learned representations. A widely known property is the translation equivariance of convolutional neural nets (CNN) [28]: through spatial or temporal parameter sharing, a shifted input leads to a shifted output. Group-equivariant CNNs extend the equivariance to rotation, mirror reflection and translation [7] by replacing the shift operation with a more general set of transformations. Other representations for building equivariance into deep nets have also been proposed, *e.g.*, the Symmetric Network [12], the Harmonic Network [57] and the Spherical CNN [8].

The aforementioned works focus on deep nets where the input are images. While related, they are not directly applicable to human pose. For example, a reflection with respect to the y-axis in the image domain corresponds to a permutation of the pixel locations, *i.e.*, swapping the pixel intensity between each pixel's reflected counterpart. In contrast, for human pose, where the input is a vector representing the human joints' spatial coordinates, a reflection corresponds to the *negation of the value* for each of the joints reflected dimension.

The input representation of deep nets for human pose is more similar to pointsets. Prior work has explored building permutation equivariant deep nets, *i.e.*, any permutation of input elements results in the same permutation of output elements. In [62, 43]. Both works utilize parameter sharing to achieve permutation equivariance. Following these works, graph nets generalize the family of permutation equivariant networks and demonstrate success on numerous applications [46, 27, 14, 13, 1, 26, 61, 31].

For human pose, equivariance to *all permutations* is too strong of a property. Recall, our aim is to build models equivariant to the chiral symmetry, which only involves *a specific permutation*, *e.g.*, the switch between left and right joints, shown in step (2) of Fig. 1.

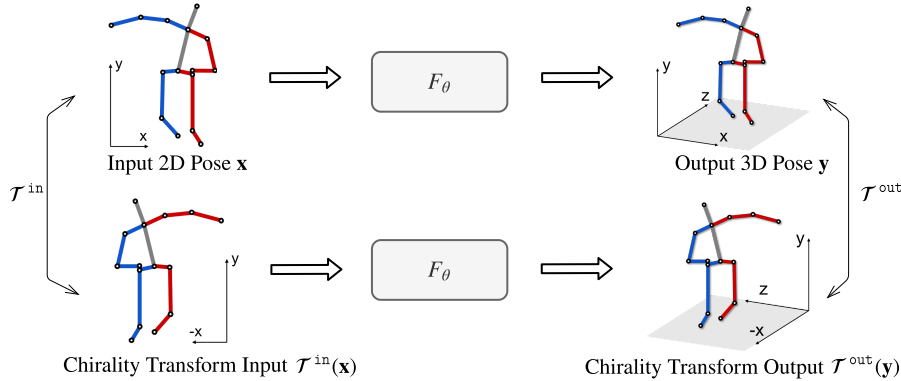

Figure 2: Illustration of chirality equivariance for the task of 2D to 3D pose estimation.

Most relevant to our approach is work by Ravanbakhsh et al. [44]. Ravanbakhsh et al. [44] explore which type of equivariance can be achieved through parameter sharing. Their approach captures one specific permutation in the pose symmetric transform, but does not capture the negation from the reflection, shown in Fig. 1 step (1). In contrast, our approach considers both operations (1) and (2) jointly, which leads to a different formulation. Lastly, to the best of our knowledge, [44] only discusses theoretically the construction of equivariant networks. In this work, we design and implement a variety of building blocks for deep nets and demonstrate the benefits on a wide range of practical applications in human pose regression tasks.

**Human pose applications.** For 3D pose estimation from images, recent approaches utilize a two-step approach: (1) 2D pose keypoints are predicted given a video; (2) 3D keypoints are estimated given 2D joint locations. The 2D to 3D estimation is formulated as a regression task via deep nets [40, 52, 35, 51, 10, 41, 59, 33, 17, 29, 42]. Capturing the temporal information is crucial and has been explored in 3D pose estimation [17, 29] as well as in action recognition [53, 20], video segmentation[18, 19] and learning object dynamics [34, 37]. Most recently, Pavllo et al. [42] propose to use temporal convolutions to better capture the temporal information for 3D pose estimation over previous RNN based methods. They also performed train and test time augmentation based on the chiral-symmetric transformation. For test time augmentation, they compute the output for both the original input and the transformed input, using the average outputs as the final prediction. In contrast to our work, we note that Pavllo et al. [42] need to transform the output of the transformed input back to the original pose. To carefully assess the benefits of chirality nets, in this work, we closely follow the experiment setup of Pavllo et al. [42].

For 2D keypoint forecasting, we follow the setup of standard temporal modeling: conditioning on past observations to predict the future. To improve temporal modeling, recent works, have utilized different sequence to sequence models for this task [34, 3, 5]. In this work, we closely follow the experiment setup of Chiu et al. [5].

For action recognition, skeleton based methods have been explored extensively recently [58, 63, 30, 48] due to robustness to illumination changes and cluttered background. Here we closely follow the experimental setup of Yan et al. [58].

# 3 Chirality Nets

In the following we first provide the problem formulation for human pose regression, before defining chirality nets, equivariance and the chirality transform. Subsequently we discuss how to develop typical layers such as the fully connected layer, the convolution, *etc*., which make up chirality nets. The Pytorch implementation and unit-tests of the proposed layers are part of the supplementary material. We have also included a short Jupyter notebook demo to illustrate the key concepts.

## 3.1 Problem Formulation

Chirality nets can be applied to regression tasks on coordinates of joints for human pose related task, *i.e.*, the input corresponds to 2D or 3D coordinates of human joints. For readability, we introduce the input and output representations for a single frame. Note that for our experiments we generalize chirality nets to multiple frames by introducing a time dimension.

We let $\mathbf{x} \in \mathbb{R}^{|J^{\text{in}}| \cdot |D^{\text{in}}|}$ denote the chirality net input, where $J^{\text{in}}$ is the set of all joints and $D^{\text{in}}$ is the dimension index set for an input coordinate. For example, $J^{\text{in}} = \{\text{'right wrist', 'right shoulder'}, \dots\}$ and $D^{\text{in}} = \{0, 1\}$, for 2D input joint coordinates. Similarly, we let $\mathbf{y} \in \mathbb{R}^{|J^{\text{out}}| \cdot |D^{\text{out}}|}$ refer to the chirality net output. Note that the dimension of the spatial coordinates at the input and output may be different, *e.g.*, prediction from 2D to 3D. Also, the number of joints may differ, *e.g.*, when mapping between different key-point sets.

For human pose regression, the task is to learn the parameters $\theta$ of a model $F_\theta$ by minimizing a loss function, $\mathcal{L}(\theta) = \sum_{(\mathbf{x}, \mathbf{y}) \in \mathcal{D}} \ell(F_\theta(\mathbf{x}), \mathbf{y})$ over the training dataset $\mathcal{D}$. Hereby, sample loss $\ell(F_\theta(\mathbf{x}), \mathbf{y})$ compares prediction $F_\theta$ to ground-truth $\mathbf{y}$.

## 3.2 Chirality Nets, Chirality Equivariance, and Chirality Transforms

Chirality nets exhibit chirality equivariance, *i.e.*, their output is transformed in a "predefined manner" given that the chirality transform is applied at the input. Note that the input and output dimensions $D^{\text{in}}$ and $D^{\text{out}}$ may differ. To define this chirality equivariance, we hence need to consider a pair of transformations, one for the input data, $\mathcal{T}^{\text{in}}$, and one for the output data, $\mathcal{T}^{\text{out}}$. The corresponding equivariance map is illustrated in Fig. 2 for the task of 2D to 3D pose estimation. Formally, we say a function $F_\theta$ is chirality equivariant w.r.t. $(\mathcal{T}^{\text{in}}, \mathcal{T}^{\text{out}})$ if

$$\mathcal{T}^{\text{out}}(F_\theta(\mathbf{x})) = F_\theta(\mathcal{T}^{\text{in}}(\mathbf{x})) \ \ \forall \mathbf{x} \in \mathbb{R}^{|J^{\text{in}}||D^{\text{in}}|}.$$

To define the chirality transform on the input data, *i.e.*, $\mathcal{T}^{\text{in}}$, we split the set of joints $J^{\text{in}}$ into *ordered tuples* of $J_{\text{l}}^{\text{in}}$, $J_{\text{r}}^{\text{in}}$, and $J_{\text{c}}^{\text{in}}$, each denoting left, right and center joints of the input. Importantly, these tuples are sorted such that the corresponding left/right joints are at corresponding positions in the tuple. We also split the dimension index set $D^{\text{in}}$ into $D_{\text{n}}^{\text{in}}$ and $D_{\text{p}}^{\text{in}} := D^{\text{in}} \backslash D_{\text{n}}^{\text{in}}$, indicating the coordinates to, or not to, negate.

For readability and without loss of generality, assume the dimensions of the input $\mathbf{x}$ follow the order of $J_{\text{l}}^{\text{in}}$, $J_{\text{r}}^{\text{in}}$, $J_{\text{c}}^{\text{in}}$, *i.e.*, $\mathbf{x} = [\mathbf{x}_{\text{l}}, \mathbf{x}_{\text{r}}, \mathbf{x}_{\text{c}}]$. Within each vector $\mathbf{x}_{(\cdot)}$, we place the coordinates in the set $D_{\text{n}}^{\text{in}}$ before the remaining ones, *i.e.*, $\mathbf{x}_{\text{l}} = [\mathbf{x}_{\text{ln}}, \mathbf{x}_{\text{lp}}]$.

Given this construction of the input $\mathbf{x}$, the reflection illustrated in step (1) of Fig. 1 is a matrix multiplication with a $(|J^{\text{in}}||D^{\text{in}}|) \times (|J^{\text{in}}||D^{\text{in}}|)$ diagonal matrix $T_{\text{neg}}^{\text{in}}$, defined as follows:

$$T_{\text{neg}}^{\text{in}} = \texttt{diag}([-\mathbf{1}_{|J_{\text{l}}^{\text{in}}| \cdot |D_{\text{n}}^{\text{in}}|}, \mathbf{1}_{|J_{\text{l}}^{\text{in}}| \cdot |D_{\text{p}}^{\text{in}}|}, -\mathbf{1}_{|J_{\text{r}}^{\text{in}}| \cdot |D_{\text{n}}^{\text{in}}|}, \mathbf{1}_{|J_{\text{r}}^{\text{in}}| \cdot |D_{\text{p}}^{\text{in}}|}, -\mathbf{1}_{|J_{\text{c}}^{\text{in}}| \cdot |D_{\text{n}}^{\text{in}}|}, \mathbf{1}_{|J_{\text{c}}^{\text{in}}| \cdot |D_{\text{p}}^{\text{in}}|}]),$$

where $\mathbf{1}_K$ indicates a vector of ones of length $K$. The switch operation illustrated in step (2) of Fig. 1 is a matrix multiplication with a permutation matrix of dimension $(|J^{\text{in}}||D^{\text{in}}|) \times (|J^{\text{in}}||D^{\text{in}}|)$, defined as follows:

$$T_{\texttt{swi}}^{\text{in}} = \begin{bmatrix} \mathbf{0} & \mathbf{I}_{|J_{\text{l}}^{\text{in}}| \cdot |D^{\text{in}}|} & \mathbf{0} \\ \mathbf{I}_{|J_{\text{l}}^{\text{in}}| \cdot |D^{\text{in}}|} & \mathbf{0} & \mathbf{0} \\ \mathbf{0} & \mathbf{0} & \mathbf{I}_{|J_{\text{c}}^{\text{in}}| \cdot |D^{\text{in}}|} \end{bmatrix},$$

where $\mathbf{I}_K$ denotes an identity matrix of size $K \times K$.

Given those matrices, the chirality transform of the input $\mathcal{T}^{\text{in}}(\mathbf{x})$ is obtained via $\mathcal{T}^{\text{in}}(\mathbf{x}) = T_{\text{neg}}^{\text{in}} T_{\texttt{swi}}^{\text{in}} \mathbf{x}$. The chirality transform of the output, $\mathcal{T}^{\text{out}}$, is defined similarly, replacing "in" with "out".

In the following, we introduce layers that satisfy the $(\mathcal{T}^{\text{in}}, \mathcal{T}^{\text{out}})$ chirality equivariance property. This enables to construct a chirality net $F_\theta$, as the composition of equivariant layers remains equivariant. Note that $(\mathcal{T}^{\text{in}}, \mathcal{T}^{\text{out}})$ chirality equivariance can be specified separately for every deep net layer which provides additional flexibility. In the following we discuss how to construct layers which satisfy chirality equivariance.

## 3.3 Chirality Layers

**Fully connected layer.** A fully connected layer performs the mapping $\mathbf{y} = f_{\text{FC}}(\mathbf{x}; W, b) := W\mathbf{x} + b$. We achieve equivariance through parameter sharing and odd symmetry:

$$W = \begin{bmatrix} \begin{bmatrix} W_{\mathtt{ln,ln}} & W_{\mathtt{ln,lp}} \\ W_{\mathtt{lp,ln}} & W_{\mathtt{lp,lp}} \end{bmatrix} & \begin{bmatrix} W_{\mathtt{ln,rn}} & W_{\mathtt{ln,rp}} \\ W_{\mathtt{lp,rn}} & W_{\mathtt{lp,rp}} \end{bmatrix} & \begin{bmatrix} W_{\mathtt{ln,cn}} & W_{\mathtt{ln,cp}} \\ W_{\mathtt{lp,cn}} & W_{\mathtt{lp,cp}} \end{bmatrix} \\ \begin{bmatrix} W_{\mathtt{ln,rn}} & -W_{\mathtt{ln,rp}} \\ -W_{\mathtt{lp,rn}} & W_{\mathtt{lp,rp}} \end{bmatrix} & \begin{bmatrix} W_{\mathtt{ln,ln}} & -W_{\mathtt{ln,lp}} \\ -W_{\mathtt{lp,ln}} & W_{\mathtt{lp,lp}} \end{bmatrix} & \begin{bmatrix} W_{\mathtt{ln,cn}} & -W_{\mathtt{ln,cp}} \\ -W_{\mathtt{lp,cn}} & W_{\mathtt{lp,cp}} \end{bmatrix} \\ \begin{bmatrix} W_{\mathtt{cn,ln}} & W_{\mathtt{cn,lp}} \\ \mathbf{0} & W_{\mathtt{cp,lp}} \end{bmatrix} & \begin{bmatrix} W_{\mathtt{cn,ln}} & -W_{\mathtt{cn,lp}} \\ \mathbf{0} & W_{\mathtt{cp,lp}} \end{bmatrix} & \begin{bmatrix} W_{\mathtt{cn,cn}} & \mathbf{0} \\ \mathbf{0} & W_{\mathtt{cp,cp}} \end{bmatrix} \end{bmatrix}, \quad b = \begin{bmatrix} \begin{bmatrix} b_{\mathtt{ln}} \\ b_{\mathtt{lp}} \end{bmatrix} \\ \begin{bmatrix} -b_{\mathtt{ln}} \\ b_{\mathtt{lp}} \end{bmatrix} \\ \begin{bmatrix} \mathbf{0} \\ b_{\mathtt{cp}} \end{bmatrix} \end{bmatrix}.$$

We color code the shared parameters using identical colors. Each $W_{(\cdot),(\cdot)}$ denotes a matrix, where the first and the second subscript characterize the dimensions of the output and the input. For example, $W_{\mathtt{ln,rp}}$ computes the output's left (l) joint's negated (n) dimensions, from the input's right (r) joint's non-negated, $i.e.$, positive (p), dimensions. Note that $W_{\mathtt{ln,rp}}$ is a matrix of dimension $|J_{\mathtt{l}}^{\mathtt{out}}| \cdot |D_{\mathtt{n}}^{\mathtt{out}}| \times |J_{\mathtt{r}}^{\mathtt{in}}| \cdot |D_{\mathtt{p}}^{\mathtt{in}}|$. We refer to this layer as the chiral fully connected layer.

**1D convolution layers [55, 28].** Pose symmetric 1D convolution layers can be based on fully connected layers. A 1D convolution is a fully connected layer with shared parameters across the time dimension, $i.e.$, at each time step the computation is the sum of fully connected layers over a window:

$$\mathbf{y}_t = \sum_\tau W_\tau \mathbf{x}_{t-\tau} + b = \sum_\tau f_{\text{FC}}(\mathbf{x}_{t-\tau}; W_\tau, b).$$

Consequently, we enforce equivariance at each time step by employing the symmetry pattern of fully connected layers at each time slice.

**Element-wise nonlinearities.** Nonlinearities are applied element-wise and do not contain parameters. These operations maintain the input dimension, therefore, $\mathcal{T}^{\text{out}}$ and $\mathcal{T}^{\text{in}}$ are identical. A nonlinearity $f$ that is an odd function, $i.e.$, $f(-x) = -f(x)$, such as tanh, hardtanh, or soft-sign satisfies the equivariance property. See the following proof:

$$\mathcal{T}^{\text{out}}(f(\mathbf{x})) = \quad T_{\text{neg}}^{\text{out}} T_{\text{swi}}^{\text{out}}(f(\mathbf{x})) \overset{\text{elementwise } f}{=} T_{\text{neg}}^{\text{out}} f(T_{\text{swi}}^{\text{out}} \mathbf{x}))$$
$$\overset{\text{odd func. } f}{=} \quad f(T_{\text{neg}}^{\text{out}} T_{\text{swi}}^{\text{out}} \mathbf{x}) = f(\mathcal{T}^{\text{in}}(\mathbf{x})) \quad \forall \mathbf{x} \in \mathbb{R}^{|J^{\text{in}}||D^{\text{in}}|}.$$

**LSTM and GRU layers [16, 6].** LSTM and GRU modules which satisfy chirality can be obtained from fully connected layers. However, naïvely setting all matrix multiplies within an LSTM to satisfy the equivariance property will not lead to an equivariant LSTM because gates are elementwise *multiplied* with the cell state. If both gate and cell preserve the negation then the product will not. Therefore, we change the weight sharing scheme for the gates. We set $D_n^{\text{out}}$ for the gates to be the empty set, $i.e.$, the gates will be invariant to negation at the input, $T_{\text{neg}}^{\text{in}}$, but still equivariant to the switch operation, $T_{\text{swi}}^{\text{in}}$. With this setup, the product of the gates and the cell's output will preserve the sign, as the gates are invariant to negation and passed through a Sigmoid to be within the range of $(0, 1)$. GRU modules are modified in the same manner.

**Batch-normalization [21].** A batch normalization layer performs an element-wise standardization, followed by an element-wise affine layer (with learnable parameters $\gamma$ and $\beta$). For $\gamma$ and $\beta$, we follow the the principle applied to fully connected layers.

Equivariance for $\mu$, and $\sigma$ is obtained by computing the mean and standard deviation on the "augmented batch" and by keeping track of its running average.

**Dropout [50].** At test time, dropout scales the input by $p$, where $p$ is the dropout probability. The equivariance property is satisfied because of the associativity property of a scalar multiplication.

### 3.4 Reduction in model parameters, FLOPS, and training/test details

**Model parameters.** Our model shares parameters between dimensions representing the left and right joints. For each layer, the number of parameters are reduced by a factor of $\frac{|(|J_l^{\text{in}}| + |J_c^{\text{in}}|) \cdot (|J_l^{\text{out}}| + |J_c^{\text{out}}|)}{|J^{\text{in}}| \cdot |J^{\text{out}}|}$. Recall $|J^{\text{in}}| = |J_l^{\text{in}}| + |J_r^{\text{in}}| + |J_c^{\text{in}}|$. The output dimension size is computed similarly.

**FLOPS.** Chirality nets also have lower FLOPS. Due to the symmetry, instead of multiplying and adding each of the elements independently, we add the symmetric values first before applying a single multiplication per symmetric pair. Concretely, consider $\mathbf{w} = [w_1, w_1]$, $\mathbf{x} = [x_1, x_2]$, and their inner product $\mathbf{w}^T \mathbf{x}$. Instead of computing $w_1 \cdot x_1 + w_1 \cdot x_2$, we exploit symmetry and use instead $w_1 \cdot (x_1 + x_2)$, which removes one multiplication operation. This is a common speed up trick used

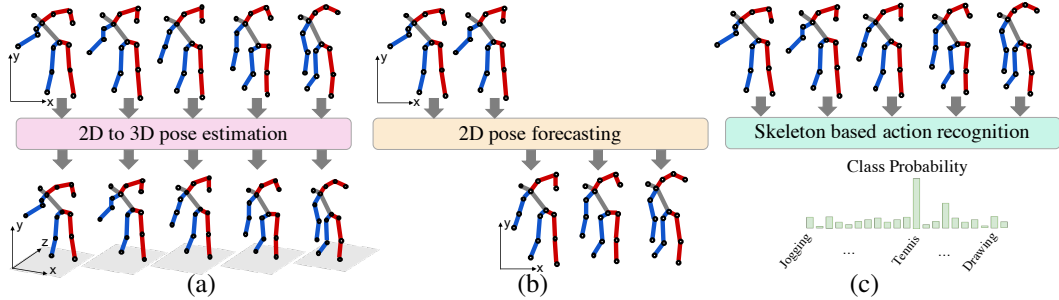

Figure 3: Illustration of pose regression tasks: (a) 2D to 3D pose estimation; (b) 2D pose forecasting; and (c) skeleton-based action recognition.

| Approach | Dir. | Disc. | Eat | Greet | Phone | Photo | Pose | Purch. | Sit | SitD. | Smoke | Wait | WalkD. | Walk | WalkT. | Avg |
|---|---|---|---|---|---|---|---|---|---|---|---|---|---|---|---|---|
| Pavlakos [41] (CVPR'18) | 48.5 | 54.4 | 54.4 | 52.0 | 59.4 | 65.3 | 49.9 | 52.9 | 65.8 | 71.1 | 56.6 | 52.9 | 60.9 | 44.7 | 47.8 | 56.2 |
| Yang [59] (CVPR'18) | 51.5 | 58.9 | 50.4 | 57.0 | 62.1 | 65.4 | 49.8 | 52.7 | 69.2 | 85.2 | 57.4 | 58.4 | **43.6** | 60.1 | 47.7 | 58.6 |
| Luvizon [33] (CVPR'18) ($\diamond$) | 49.2 | 51.6 | 47.6 | 50.5 | 51.8 | 60.3 | 48.5 | 51.7 | 61.5 | 70.9 | 53.7 | 48.9 | 57.9 | 44.4 | 48.9 | 53.2 |
| Hossain [17] (ECCV'18)(†, $\diamond$) | 48.4 | 50.7 | 57.2 | 55.2 | 63.1 | 72.6 | 53.0 | 51.7 | 66.1 | 80.9 | 59.0 | 57.3 | 62.4 | 46.6 | 49.6 | 58.3 |
| Lee [29] (ECCV'18)(†, $\diamond$) | **40.2** | 49.2 | 47.8 | 52.6 | 50.1 | 75.0 | 50.2 | **43.0** | **55.8** | 73.9 | 54.1 | 55.6 | 58.2 | 43.3 | 43.3 | 52.8 |
| Pavllo [42] (CVPR'19) | 47.1 | 50.6 | 49.0 | 51.8 | 53.6 | 61.4 | 49.4 | 47.4 | 59.3 | 67.4 | 52.4 | 49.5 | 55.3 | 39.5 | 42.7 | 51.8 |
| Pavllo [42] (CVPR'19)(†) | 45.9 | 47.5 | 44.3 | _46.4_ | 50.0 | 56.9 | 45.6 | 44.6 | 58.8 | 66.8 | 47.9 | 44.7 | 49.7 | 33.1 | 34.0 | 47.7 |
| Pavllo [42] (CVPR'19)(†, ‡) | 45.2 | 46.7 | **43.3** | **45.6** | **48.1** | **55.1** | 44.6 | 44.3 | _57.3_ | 65.8 | **47.1** | 44.0 | 49.0 | 32.8 | 33.9 | _46.8_ |
| Ours, single-frame | 47.4 | 49.9 | 47.4 | 51.1 | 53.8 | 61.2 | 48.3 | 45.9 | 60.4 | 67.1 | 52.0 | 48.6 | 54.6 | 40.1 | 43.0 | 51.4 |
| Ours (†) | _44.8_ | **46.1** | **43.3** | _46.4_ | _49.0_ | _55.2_ | 44.6 | **44.0** | 58.3 | **62.7** | **47.1** | 43.9 | _48.6_ | **32.7** | **33.3** | **46.7** |

Table 1: Results on the Human3.6M dataset: reconstruction error using Protocol 1 (MPJPE) in mm. The best result is boldface and the second best is underlined. † indicates temporal models, $\diamond$ uses ground-truth bounding box, and ‡ indicates test-time augmentation.

in symmetric FIR filters [38, 60]. The number of multiplications reduces by a factor of $\frac{|J_l^{\text{in}}| + |J_c^{\text{in}}|}{|J^{\text{in}}|}$. Additionally, baseline models utilize test-time augmentation, which requires two forward passes through the network for each input, whereas the proposed nets only use a single forward pass.

**Training and test details.** During training it is important to apply the chirality transform for data-augmentation, *i.e.*, with 50% probability we apply $\mathcal{T}^{\text{in}}$ and $\mathcal{T}^{\text{out}}$ to input and label. This ensures that the mini-batch statistics match our assumption on the chirality, *i.e.*, poses that form a chiral pair are both valid, which is important for the batch-normalization layer. Moreover, during training we use a standard dropout layer. While we could impose dropped units to be chiral equivariant, we found this lead to over-fitting in practice. This is expected as imposing chirality on the added noise reduces the randomness. Importantly, during test no data-augmentation is performed and a single forward pass is sufficient to obtain an 'averaged' result.

## 4 Experiments

We evaluate our approach on a variety of tasks, including 2D to 3D pose estimation, 2D pose forecasting, and skeleton based action recognition. For each task, we describe the dataset, metric, and implementation before discussing the results.

### 4.1 2D to 3D pose estimation

**Task.** 3D human pose estimation can be decoupled into the tasks of 2D keypoint detection and 2D to 3D pose estimation. We focus on the latter task, *i.e.*, given a sequence of 2D keypoints, the task is to estimate the corresponding 3D human pose. See Fig. 3 (a) for an illustration.

**Dataset and metric.** We evaluate on two standard datasets, the Human3.6M [22] and the HumanEva-I [49]. Human3.6M is a large scale dataset of human motion with 3.6 million video frames. The dataset consists of 11 subjects performing 15 different actions. Following prior work [40, 52, 35, 51, 33, 42], each human pose is represented by a 17-joint skeleton. We use the same train and test subject splits. HumanEva-I is a smaller dataset consisting of four subjects and six actions. To be consistent with prior work [41, 29, 42], we use the same train and test splits evaluated over the actions of (walk, jog, and box). For both of these datasets, we consider the setting where we train one model for all actions.

We report the two standard metrics used in prior work: Protocol 1 (MPJPE) which is the mean per-joint position error between the prediction and ground-truth [35, 40, 42] and Protocol 2 (P-MPJPE) which is the error, after alignment, between the prediction and ground-truth [35, 51, 17, 42].

| App. | Walk | | | Jog | | | Box | | | Avg. |
|---|---|---|---|---|---|---|---|---|---|---|
| | S1 | S2 | S3 | S1 | S2 | S3 | S1 | S2 | S3 | - |
| Pavlakos [40] | 22.3 | 19.5 | 29.7 | 28.9 | 21.9 | 23.8 | – | – | – | – |
| Pavlakos [41] | 18.8 | 12.7 | **29.2** | 23.5 | 15.4 | 14.5 | – | – | – | – |
| Lee [29] | 18.6 | 19.9 | 30.5 | 25.7 | 16.8 | 17.7 | 42.8 | 48.1 | 53.4 | – |
| Pavllo [42] | 14.1 | 10.4 | 46.8 | 21.1 | 13.3 | 14.0 | 23.8 | 34.5 | 32.3 | 31.1 |
| Pavllo [42] (‡) | **13.9** | **10.2** | 46.6 | **20.9** | 13.1 | 13.8 | 23.8 | 33.7 | 32.0 | 30.8 |
| Ours | 15.2 | 10.3 | 47.0 | 21.8 | **13.1** | **13.7** | **22.8** | **31.8** | **31.0** | **30.6** |

Table 2: Results on HumanEva-I for multi-action (MA) models reported in Protocol 2 (P-MPJPE), lower the better. ‡ indicates test time augmentation.

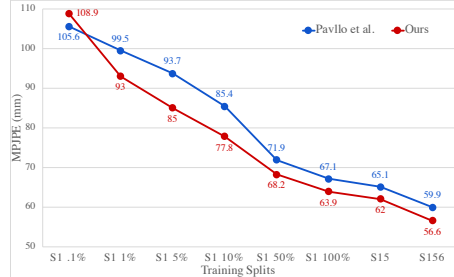

Figure 4: Comparisons between our approach and [42] in limited data settings evaluated using Protocol 1 on Human3.6M.

| Approach | Prediction Steps | | | | | | | | | | | | | | | | Avg. |
|---|---|---|---|---|---|---|---|---|---|---|---|---|---|---|---|---|---|
| | 1 | 2 | 3 | 4 | 5 | 6 | 7 | 8 | 9 | 10 | 11 | 12 | 13 | 14 | 15 | 16 | - |
| Residual [34] (CVPR'17) | 82.4 | 68.3 | 58.5 | 50.9 | 44.7 | 40.0 | 36.4 | 33.4 | 31.3 | 29.5 | 28.3 | 27.3 | 26.4 | 25.7 | 25.0 | 24.5 | 39.5 |
| 3D-PFNet [3](CVPR'17) | 79.2 | 60.0 | 49.0 | 43.9 | 41.5 | 40.3 | 39.8 | 39.7 | 40.1 | 40.5 | 41.1 | 41.6 | 42.3 | 42.9 | 43.2 | 43.3 | 45.5 |
| TP-RNN [5] (WACV'19) | 84.5 | 72.0 | 64.8 | 60.3 | 57.2 | 55.0 | 53.4 | 52.1 | 50.9 | 50.0 | 49.3 | 48.7 | 48.3 | 47.9 | 47.6 | 47.3 | 55.6 |
| Baseline w/o aug. | 87.3 | 75.7 | 68.5 | 64.0 | 61.0 | 59.1 | **57.6** | 56.3 | 55.4 | 54.9 | 54.5 | 54.4 | 54.5 | 54.5 | 54.6 | **54.7** | 60.4 |
| Baseline w/ aug. | 86.9 | 75.2 | 67.9 | 63.5 | 60.4 | 58.4 | 57.0 | 55.8 | 55.1 | 54.5 | 54.1 | 54.0 | 53.9 | 53.9 | 54.0 | 54.0 | 59.9 |
| Baseline w/ aug.(‡) | 87.0 | 75.5 | 68.4 | 64.1 | 61.0 | 59.1 | 57.5 | 56.3 | 55.5 | 55.0 | **54.7** | **54.7** | **54.6** | **54.7** | **54.7** | **54.7** | 60.5 |
| Ours | **87.5** | **77.0** | **68.7** | **64.2** | **61.2** | **59.2** | **57.6** | **56.5** | **55.7** | **55.1** | **54.7** | 54.6 | 54.4 | 54.5 | 54.5 | 54.5 | **60.6** |

Table 3: Results on Penn action dataset, performance reported in terms of PCK@0.05 (higher the better). (‡) indicates using test time augmentation.

**Implementation details.** Our model follows the supervised training procedure and network design of Pavllo et al. [42]. Our network is the identical temporal convolutional network architecture, where each layer is replaced with its chiral version, *i.e.*, 1D dilated convolution, batch-normalization, and dropout layers. We also replace ReLU non-linearities with Tanh to achieve equivariance. No additional architecture changes were made. For Human3.6M, we use 2D keypoints extracted from CPN [4] with Mask R-CNN [15] bounding boxes released by Pavllo et al. [42]. For HumanEva-I, we use the 2D keypoint detections from Mask R-CNN released by Pavllo et al. [42].

**Results.** In Tab. 1, we report the performance on the Human3.6M data using Protocol 1 (MPJPE). Our approach outperforms the state-of-the-art [42] which uses test-time augmentation by 0.1 mm in overall average and achieves the best results in eight out of fifteen sub-categories. For the single-frame models, we observe a more significant reduction in error of 0.4 mm over [42] with test time augmentation. Additionally, when comparing without test-time augmentation, our approach outperforms by 1 mm. We note that, test-time augmentation employed by Pavllo et al. [42] involves running the network twice for each input. In contrast, our approach only requires a single forward pass.

Next, on HumanEva-I dataset, we also observed an increase in performance using Protocol 1. On average, our approach achieves a 32.2mm error. This is a 0.8mm decrease over the current state-of-the-art of 33.0mm [42] and a 1.1mm decrease over [42] without test-time augmentation of 33.3mm.

We also performed evaluation using Protocol 2 (P-MPJPE). On Human3.6M we observe that our approach performs worse than Pavllo et al. [42] by 0.3mm. We note that the loss function is chosen to optimize Protocol 1, therefore our models are performing better at what they are optimized for. In Tab. 2, we report the performance on HumanEva-I using Protocol 2 (P-MPJPE). Our model achieves a 0.2 mm reduction in error over Pavllo et al. [42] on average. Most of the gain is obtained for the boxing action, possibly due to the symmetric nature of the movement.

**Limited data settings.** A benefit of fewer model parameters is the potential to obtain better models with less data. To confirm this, we perform experiments by varying the amount of training data, starting from 0.1% of subject 1 (S1) to using three subjects S1, S5, S6. The results with comparison to [42] are shown in Fig. 4. We observe that our approach consistently out-performs [42] in this low resource settings, except at S1 0.1%. For the reported numbers, we use a batch-size of 64, and all other hyper-parameters are identical between the models. If we further decrease the batch-size to 32 for S1 0.1%, our approach improves to 100.4mm where [42] improves to 102.3mm.

## 4.2  2D pose forecasting

**Task.** 2D pose forecasting is the pose regression task of predicting the future human pose, represented in 2D keypoints, given present and past human pose. See Fig. 3 (b) for an illustration.

**Dataset and metric.** We evaluate on the Penn Action dataset [64]. The dataset consists of 2236 videos with 15 actions. Each frame is annotated with 2D keypoints of 13 human joints. We use the same train and test split as in [3, 5]. Following Chiu et al. [5] we consider initial velocity as being part of the input and a single model is used for all actions. For a fair comparison with prior work, we report the 'Percentage of Correct Keypoint' metric with a 0.05 threshold (PCK@0.05), which assesses the accuracy of the predicted keypoints. A predicted keypoint is considered correct if it is within a 0.05 radius of the ground-truth when considering normalized distance.

**Implementation details.** Our non-chiral equivariant baseline model is a sequence-to-sequence model based on [34]. We made several modifications to match the hyperparameters in [5], *i.e.*, we used StackedRNN [39] with 2 layers and added dropout layers. Additionally, we utilize teacher forcing [56] during training, while prior work did not. We find this to stabilize training and enable the use of the Adam [25, 45] optimizer without diverging. We performed data augmentation via the chirality transform, *i.e.*, with 0.5 probability we apply $\mathcal{T}^{\mathtt{in}}$ and $\mathcal{T}^{\mathtt{out}}$ to the input and the ground-truth correspondingly. For our pose symmetric model, we replaced all the non-symmetric layers, *e.g.*, fully connected layers and LSTM cells with their corresponding chiral version.

**Results.** In Tab. 3, we report the performance of our models and the state-of-the-art. The baseline model without augmentation outperforms the state-of-the-art [5]. The gain comes from the use of Stacked-LSTM and teacher forcing during training. With additional train and test time data-augmentation, our baseline model further improves. In addition our pose symmetric model outperforms the baseline, in terms of average PCK@0.05. We observe more significant improvements for the first ten prediction steps.

### 4.3 Skeleton based action recognition

**Task.** Skeleton based action recognition aims at predicting human action based on skeleton sequences. See Fig. 3 (c) for an illustration.

| Approach | Top-1 | Top-5 |
|---|---|---|
| Feature Encoding [11] | 14.9% | 25.8% |
| Deep LSTM [47] | 16.4% | 35.3% |
| Temporal-Conv [24] | 20.3% | 40.0% |
| ST-GCN [58] | 30.7% | 52.8% |
| Ours-Conv | 30.8% | 52.6% |
| Ours-Conv-Chiral | **30.9%** | **53.0%** |

Table 4: Results of the skeleton based action recognition baselines on the Kinetics-400 dataset [23] reported in Top-1 and Top-5 accuracy.

**Dataset and metric.** We use the Kinetics-400 dataset [23] in our experiments. The dataset contains 400 action classes and 306,245 clips in total. Following the experimental setup by [58], we use OpenPose [2] to locate the 18 human body joints. Each joint is represented as $(x, y, c)$, where $x$ and $y$ are the 2D coordinates of the joint and $c$ is the confidence score of the joint given by OpenPose. Following [23], we report the classification accuracy at top-1 and top-5.

**Implementation details.** Our baseline model, 'Ours-Conv,' follows 'Temporal-Conv' [24], modified to have not only temporal convolution but also spatial convolution. The temporal convolution considers the intra-frame information while the spatial convolution considers the inter-frame information. For the recognition task, we need chiral invariance, *i.e.*, a chiral pair should be classified as the same action class. To this end, we use a chiral invariance layer where we let both $J_{\mathtt{r}}^{\mathtt{out}}$, $J_{\mathtt{l}}^{\mathtt{out}}$ as well as $D_{\mathtt{n}}^{\mathtt{out}}$ be empty sets, which means there are no left and right joints but only center joints and there is no dimension that will be negated in the output of the layer after applying chirality transform. Note that the chirality transform exchanges the left and right joints and negates the dimensions in the dimension index set $D_{\mathtt{n}}^{\mathtt{out}}$. Given $J_{\mathtt{r}}^{\mathtt{out}}$, $J_{\mathtt{l}}^{\mathtt{out}}$ and $D_{\mathtt{n}}^{\mathtt{out}}$ are all empty, it's trivial that the output will be chiral invariant. For the chiral invariance model, 'Ours-Conv-Chiral,' we replace all the non-symmetric layers before the chiral invariance layer with their corresponding chiral equivariance version. All the layers after the chiral invariance layer remain identical to the 'Ours-Conv' model. There are in total 10 layers of spatial and temporal convolution and we put the chiral invariance layer at the fourth layer. We use the SGD optimizer with a momentum of $0.9$ as in [58].

**Results.** In Tab. 4, we report the action recognition performance of our model and the skeleton-based approaches. We observe that the baseline model 'Ours-Conv' performs on par with ST-GCN [58] and the chiral invariant model, 'Ours-Conv-Chiral' outperforms both ST-GCN and Ours-Conv on Top-1 and Top-5 accuracy, achieving the state-of-the-art performance on the Kinetics-400 dataset among skeleton based action recognition methods.

## 5 Conclusion

We introduce chirality equivariance for pose regression tasks and develop deep net layers that satisfy this property. Through parameter sharing and odd/even symmetry, we design equivariant versions of

commonly used layers in deep nets, including fully connected, 1D convolution, LSTM/GRU cells, and batch normalization layers. With these equivariant layers at hand, we build Chirality Nets, which guarantee equivariance from the input to the output. Our models naturally lead to a reduction in trainable parameters and computation due to symmetry. Our experimental results on three human pose regression tasks over four datasets demonstrate state-of-the-art performance and the wide practical impact of the proposed layers.

**Acknowledgments:** This work is supported in part by NSF under Grant No. 1718221 and MRI #1725729, UIUC, Samsung, 3M, Cisco Systems Inc. (Gift Award CG 1377144) and Adobe. We thank NVIDIA for providing GPUs used for this work and Cisco for access to the Arcetri cluster. RY is supported by a Google PhD Fellowship.

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
