[Supplementary Material 1]



# Supplementary material: Pose Symmetric Network for Human Pose Regression

## A  Code and Test Cases

In the supplemental materials, we have included Pytorch implementation of the proposed layers. Each layer also comes with unit-tests validating the chirality-equivaraince. Please read the `README.md` for directory structures, usage and required dependencies. There is also a Jupyter notebook and it's HTML output visualizing the concepts introduced in the paper.

## B  Additional Description for Equivariant Layers

### B.1  Equivariant fully connected layers

Recall, we achieve equivariance through parameter sharing and odd symmetry.

A fully connected layer performs the mapping $\mathbf{y} = f_{\text{FC}}(\mathbf{x}; W, b) := W\mathbf{x} + b$. Recall, we achieve equivariance through parameter sharing and odd symmetry:

$$W = \begin{bmatrix} \begin{bmatrix} W_{\text{ln,ln}} & W_{\text{ln,lp}} \\ W_{\text{lp,ln}} & W_{\text{lp,lp}} \end{bmatrix} & \begin{bmatrix} W_{\text{ln,rn}} & W_{\text{ln,rp}} \\ W_{\text{lp,rn}} & W_{\text{lp,rp}} \end{bmatrix} & \begin{bmatrix} W_{\text{ln,cn}} & W_{\text{ln,cp}} \\ W_{\text{lp,cn}} & W_{\text{lp,cp}} \end{bmatrix} \\ \begin{bmatrix} W_{\text{ln,rn}} & -W_{\text{ln,rp}} \\ -W_{\text{lp,rn}} & W_{\text{lp,rp}} \end{bmatrix} & \begin{bmatrix} W_{\text{ln,ln}} & -W_{\text{ln,lp}} \\ -W_{\text{lp,ln}} & W_{\text{lp,lp}} \end{bmatrix} & \begin{bmatrix} W_{\text{ln,cn}} & -W_{\text{ln,cp}} \\ -W_{\text{lp,cn}} & W_{\text{lp,cp}} \end{bmatrix} \\ \begin{bmatrix} W_{\text{cn,ln}} & W_{\text{cn,lp}} \\ \mathbf{0} & W_{\text{cp,lp}} \end{bmatrix} & \begin{bmatrix} W_{\text{cn,ln}} & -W_{\text{cn,lp}} \\ \mathbf{0} & W_{\text{cp,lp}} \end{bmatrix} & \begin{bmatrix} W_{\text{cn,cn}} & \mathbf{0} \\ \mathbf{0} & W_{\text{cp,cp}} \end{bmatrix} \end{bmatrix}, \quad b = \begin{bmatrix} \begin{bmatrix} b_{\text{ln}} \\ b_{\text{lp}} \end{bmatrix} \\ \begin{bmatrix} -b_{\text{ln}} \\ b_{\text{lp}} \end{bmatrix} \\ \begin{bmatrix} \mathbf{0} \\ b_{\text{cp}} \end{bmatrix} \end{bmatrix}$$

Here, we prove that the design is chiral-equivariant. Through multiplying out the matrices, we can show $W\mathcal{T}(\mathbf{x}) + b = \mathcal{T}(W\mathbf{x} + b)$, as follows:

*Proof:*

$\mathbf{x} = \begin{bmatrix} x_{\text{ln}} & x_{\text{lp}} & x_{\text{rn}} & x_{\text{rp}} & x_{\text{cn}} & x_{\text{cp}} \end{bmatrix}^T$ then $\mathcal{T}(\mathbf{x}) = \begin{bmatrix} -x_{\text{rn}} & x_{\text{rp}} & -x_{\text{ln}} & x_{\text{lp}} & -x_{\text{cn}} & x_{\text{cp}} \end{bmatrix}^T$

With linear algebra,

$$W\mathbf{x}+b = \begin{bmatrix} W_{\text{ln,ln}}(x_{\text{ln}}) + W_{\text{ln,lp}}(x_{\text{lp}}) + W_{\text{ln,rn}}(x_{\text{rn}}) + W_{\text{ln,rp}}(x_{\text{rp}}) + W_{\text{ln,cn}}(x_{\text{cn}}) + W_{\text{ln,cp}}(x_{\text{cp}}) + b_{\text{ln}} \\ W_{\text{lp,ln}}(x_{\text{ln}}) + W_{\text{lp,lp}}(x_{\text{lp}}) + W_{\text{lp,rn}}(x_{\text{rn}}) + W_{\text{lp,rp}}(x_{\text{rp}}) + W_{\text{lp,cn}}(x_{\text{cn}}) + W_{\text{lp,cp}}(x_{\text{cp}}) + b_{\text{lp}} \\ W_{\text{ln,rn}}(x_{\text{ln}}) - W_{\text{ln,rp}}(x_{\text{lp}}) + W_{\text{ln,ln}}(x_{\text{rn}}) - W_{\text{ln,lp}}(x_{\text{rp}}) + W_{\text{ln,cn}}(x_{\text{cn}}) - W_{\text{ln,cp}}(x_{\text{cp}}) - b_{\text{ln}} \\ -W_{\text{lp,rn}}(x_{\text{ln}}) + W_{\text{lp,rp}}(x_{\text{lp}}) - W_{\text{,lp,ln}}(x_{\text{rn}}) + W_{\text{lp,lp}}(x_{\text{rp}}) - W_{\text{lp,cn}}(x_{\text{cn}}) + W_{\text{lp,cp}}(x_{\text{cp}}) + b_{\text{lp}} \\ W_{\text{cn,ln}}(x_{\text{ln}}) + W_{\text{cn,lp}}(x_{\text{lp}}) + W_{\text{cn,ln}}(x_{\text{rn}}) - W_{\text{cn,lp}}(x_{\text{rp}}) + W_{\text{cn,cn}}(x_{\text{cn}}) + \mathbf{0} \cdot (x_{\text{cp}}) + \mathbf{0} \\ \mathbf{0} \cdot (x_{\text{ln}}) + W_{\text{cp,lp}}(x_{\text{lp}}) + \mathbf{0} \cdot (x_{\text{rn}}) + W_{\text{cp,lp}}(x_{\text{rp}}) + \mathbf{0} \cdot (x_{\text{cn}}) + W_{\text{cp,cp}}(x_{\text{cp}}) + b_{\text{cp}} \end{bmatrix}$$

$$\mathcal{T}(W\mathbf{x}+b) = \begin{bmatrix} -W_{\text{ln,rn}}(x_{\text{ln}}) + W_{\text{ln,rp}}(x_{\text{lp}}) - W_{\text{ln,ln}}(x_{\text{rn}}) + W_{\text{ln,lp}}(x_{\text{rp}}) - W_{\text{ln,cn}}(x_{\text{cn}}) + W_{\text{ln,cp}}(x_{\text{cp}}) + b_{\text{ln}} \\ -W_{\text{lp,rn}}(x_{\text{ln}}) + W_{\text{lp,rp}}(x_{\text{lp}}) - W_{\text{,lp,ln}}(x_{\text{rn}}) + W_{\text{lp,lp}}(x_{\text{rp}}) - W_{\text{lp,cn}}(x_{\text{cn}}) + W_{\text{lp,cp}}(x_{\text{cp}}) + b_{\text{lp}} \\ -W_{\text{ln,ln}}(x_{\text{ln}}) - W_{\text{ln,lp}}(x_{\text{lp}}) - W_{\text{ln,rn}}(x_{\text{rn}}) - W_{\text{ln,rp}}(x_{\text{rp}}) - W_{\text{ln,cn}}(x_{\text{cn}}) - W_{\text{ln,cp}}(x_{\text{cp}}) - b_{\text{ln}} \\ W_{\text{lp,ln}}(x_{\text{ln}}) + W_{\text{lp,lp}}(x_{\text{lp}}) + W_{\text{lp,rn}}(x_{\text{rn}}) + W_{\text{lp,rp}}(x_{\text{rp}}) + W_{\text{lp,cn}}(x_{\text{cn}}) + W_{\text{lp,cp}}(x_{\text{cp}}) + b_{\text{lp}} \\ -W_{\text{cn,ln}}(x_{\text{ln}}) - W_{\text{cn,lp}}(x_{\text{lp}}) - W_{\text{cn,ln}}(x_{\text{rn}}) + W_{\text{cn,lp}}(x_{\text{rp}}) - W_{\text{cn,cn}}(x_{\text{cn}}) - \mathbf{0} \cdot (x_{\text{cp}}) - \mathbf{0} \\ \mathbf{0} \cdot (x_{\text{ln}}) + W_{\text{cp,lp}}(x_{\text{lp}}) + \mathbf{0} \cdot (x_{\text{rn}}) + W_{\text{cp,lp}}(x_{\text{rp}}) + \mathbf{0} \cdot (x_{\text{cn}}) + W_{\text{cp,cp}}(x_{\text{cp}}) + b_{\text{cp}} \end{bmatrix}$$

$$W\mathcal{T}(\mathbf{x})+b = \begin{bmatrix} W_{\text{ln,ln}}(-x_{\text{rn}}) + W_{\text{ln,lp}}(x_{\text{rp}}) + W_{\text{ln,rn}}(-x_{\text{ln}}) + W_{\text{ln,rp}}(x_{\text{lp}}) + W_{\text{ln,cn}}(-x_{\text{cn}}) + W_{\text{ln,cp}}(x_{\text{cp}}) + b_{\text{ln}} \\ W_{\text{lp,ln}}(-x_{\text{rn}}) + W_{\text{lp,lp}}(x_{\text{rp}}) + W_{\text{lp,rn}}(-x_{\text{ln}}) + W_{\text{lp,rp}}(x_{\text{lp}}) + W_{\text{lp,cn}}(-x_{\text{cn}}) + W_{\text{lp,cp}}(x_{\text{cp}}) + b_{\text{lp}} \\ W_{\text{ln,rn}}(-x_{\text{rn}}) - W_{\text{ln,rp}}(x_{\text{rp}}) + W_{\text{ln,ln}}(-x_{\text{ln}}) - W_{\text{ln,lp}}(x_{\text{lp}}) + W_{\text{ln,cn}}(-x_{\text{cn}}) - W_{\text{ln,cp}}(x_{\text{cp}}) - b_{\text{ln}} \\ -W_{\text{lp,rn}}(-x_{\text{rn}}) + W_{\text{lp,rp}}(x_{\text{rp}}) - W_{\text{,lp,ln}}(-x_{\text{ln}}) + W_{\text{lp,lp}}(x_{\text{lp}}) - W_{\text{lp,cn}}(-x_{\text{cn}}) + W_{\text{lp,cp}}(x_{\text{cp}}) + b_{\text{lp}} \\ W_{\text{cn,ln}}(-x_{\text{rn}}) + W_{\text{cn,lp}}(x_{\text{rp}}) + W_{\text{cn,ln}}(-x_{\text{ln}}) - W_{\text{cn,lp}}(x_{\text{lp}}) + W_{\text{cn,cn}}(-x_{\text{cn}}) + \mathbf{0} \cdot (x_{\text{cp}}) + \mathbf{0} \\ \mathbf{0} \cdot (-x_{\text{rn}}) + W_{\text{cp,lp}}(x_{\text{rp}}) + \mathbf{0} \cdot (-x_{\text{ln}}) + W_{\text{cp,lp}}(x_{\text{lp}}) + \mathbf{0} \cdot (-x_{\text{cn}}) + W_{\text{cp,cp}}(x_{\text{cp}}) + b_{\text{cp}} \end{bmatrix}$$

observe that $W\mathcal{T}(\mathbf{x}) + b = \mathcal{T}(W\mathbf{x} + b)$, which proves the claim. $\square$

### B.2  Equivariant 1D convolution layers

**1D convolution layers [48, 24].**  Pose symmetric 1D convolution layers can be based on fully connected layers. A 1D convolution is a fully connected layer with shared parameters across the time

dimension, *i.e.*, at each time step the computation is the sum of fully connected layers over a window:

$$\mathbf{y}_t = \sum_\tau W_\tau \mathbf{x}_{t-\tau} + b = \sum_\tau f_{\text{FC}}(\mathbf{x}_{t-\tau}; W_\tau, b).$$

Consequently, we enforce equivariance at each time step by employing the symmetry pattern of fully connected layers at each time slice.

$$W_\tau = \begin{bmatrix} \begin{bmatrix} W_{\text{ln,ln},\tau} & W_{\text{ln,lp},\tau} \\ W_{\text{lp,ln},\tau} & W_{\text{lp,lp},\tau} \end{bmatrix} & \begin{bmatrix} W_{\text{ln,rn},\tau} & W_{\text{ln,rp},\tau} \\ W_{\text{lp,rn},\tau} & W_{\text{lp,rp},\tau} \end{bmatrix} & \begin{bmatrix} W_{\text{ln,cn},\tau} & W_{\text{ln,cp},\tau} \\ W_{\text{lp,cn},\tau} & W_{\text{lp,cp},\tau} \end{bmatrix} \\ \begin{bmatrix} W_{\text{ln,rn},\tau} & -W_{\text{ln,rp},\tau} \\ -W_{\text{lp,rn},\tau} & W_{\text{lp,rp},\tau} \end{bmatrix} & \begin{bmatrix} W_{\text{ln,ln},\tau} & -W_{\text{ln,lp},\tau} \\ -W_{\text{lp,ln},\tau} & W_{\text{lp,lp},\tau} \end{bmatrix} & \begin{bmatrix} W_{\text{ln,cn},\tau} & -W_{\text{ln,cp},\tau} \\ -W_{\text{lp,cn},\tau} & W_{\text{lp,cp},\tau} \end{bmatrix} \\ \begin{bmatrix} W_{\text{cn,ln},\tau} & W_{\text{cn,lp},\tau} \\ \mathbf{0} & W_{\text{cp,lp},\tau} \end{bmatrix} & \begin{bmatrix} W_{\text{cn,ln},\tau} & -W_{\text{cn,lp},\tau} \\ \mathbf{0} & W_{\text{cp,lp},\tau} \end{bmatrix} & \begin{bmatrix} W_{\text{cn,cn},\tau} & \mathbf{0} \\ \mathbf{0} & W_{\text{cp,cp},\tau} \end{bmatrix} \end{bmatrix},$$

for all $\tau$. The bias of a 1D convolution is identical to that of a fully connected layer, *i.e.*, the same bias is added for each time step. Hence the same parameter sharing is used.

### B.3 Equivariant LSTM and GRU layers

LSTM and GRU modules which satisfy chirality can be obtained from fully connected layers. However, naïvely setting all matrix multiplies within an LSTM to satisfy the equivariance property will not lead to an equivariant LSTM because gates are elementwise *multiplied* with the cell state. If both gate and cell preserve the negation then the product will not. Therefore, we change the weight sharing scheme for the gates. We set $D_n^{\text{out}}$ for the gates to be the empty set, *i.e.*, the gates will be invariant to negation at the input, $T_{\text{neg}}^{\text{in}}$, but still equivariant to the switch operation, $T_{\text{swi}}^{\text{in}}$. With this setup, the product of the gates and the cell's output will preserve the sign, as the gates are invariant to negation and passed through a Sigmoid to be within the range of $(0, 1)$. GRU modules are modified in the same manner.

More formally, the computation in an LSTM module are as follows:

$$\begin{aligned} i_t &= \sigma(W^{\text{ii}}x_t + b^{\text{ii}} + W^{\text{hi}}h_{(t-1)} + b^{\text{hi}}) & \text{(Input Gate)} \\ o_t &= \sigma(W^{\text{io}}x_t + b^{\text{io}} + W^{\text{ho}}h_{(t-1)} + b^{\text{ho}}) & \text{(Output Gate)} \\ f_t &= \sigma(W^{\text{if}}x_t + b^{\text{if}} + W^{\text{hf}}h_{(t-1)} + b^{\text{hf}}) & \text{(Forget Gate)} \\ g_t &= \tanh(W^{\text{ig}}x_t + b^{\text{ig}} + W^{\text{hg}}h_{(t-1)} + b^{\text{hg}}) & \text{(Cell State)} \\ c_t &= f_t \cdot c_{(t-1)} + i_t \cdot g_t \\ h_t &= o_t \cdot \tanh(c_t) & \text{(Recurrent State)} \end{aligned},$$

where $\sigma$ denotes an element-wise sigmoid non-linearity.

Observe that the LSTM operations consist of fully connected layers. For the cell state's parameters, *e.g.*, $W^{\text{ig}}, W^{\text{hg}}, b^{\text{ig}}, b^{\text{hg}}$, we follow the weight sharing scheme discussed for fully connected layers.

Due the to multiplication in the cell state, we redesigned the parameter sharing for the input, output and forget gate, to be invariant to $T_{\text{neg}}^{\text{in}}$, by setting $D_n^{\text{out}}$ to be the empty set: no negation is needed for all dimension. This results in the following parameter sharing scheme for the parameters $W^{\text{ii}}, b^{\text{ii}}, W^{\text{hi}}, b^{\text{hi}}, W^{\text{io}}, b^{\text{io}}, W^{\text{ho}}, b^{\text{ho}}, W^{\text{if}}, b^{\text{if}}, W^{\text{hf}}, b^{\text{hf}}$:

$$W = \begin{bmatrix} [W_{\text{1p,ln}} & W_{\text{1p,lp}}] & [W_{\text{1p,rn}} & W_{\text{1p,rp}}] & [W_{\text{1p,cn}} & W_{\text{1p,cp}}] \\ [-W_{\text{1p,rn}} & W_{\text{1p,rp}}] & [-W_{\text{1p,ln}} & W_{\text{1p,lp}}] & [-W_{\text{1p,cn}} & W_{\text{1p,cp}}] \\ [\mathbf{0} & W_{\text{cp,lp}}] & [\mathbf{0} & W_{\text{cp,lp}}] & [\mathbf{0} & W_{\text{cp,cp}}] \end{bmatrix}, b = \begin{bmatrix} [b_{\text{1p}}] \\ [b_{\text{1p}}] \\ [b_{\text{cp}}] \end{bmatrix}.$$

This LSTM is chirality equivariant, as the computation of the cell state is equivariant. Other computations are linear combinations of chirality equivariant operations, which remains equivariant. We note that the chirality equivariant GRU module is modified by following the same sharing scheme for the gates.

### B.4 Equivariant batch-norm layers

A batch normalization layer performs an element-wise standardization, followed by an element-wise affine layer (with learnable parameters $\gamma$ and $\beta$):

$$\mathbf{y} = f_{\text{BN}}(\mathbf{x}) := \gamma \cdot \frac{\mathbf{x} - \mu}{\sqrt{\sigma^2 + \epsilon}} + \beta.$$

| App. | Walk | | | Jog | | | Box | | | Avg. |
|---|---|---|---|---|---|---|---|---|---|---|
| | S1 | S2 | S3 | S1 | S2 | S3 | S1 | S2 | S3 | - |
| Pavllo [36] | 17.6 | 12.5 | 37.6 | 28.1 | 19.1 | 19.2 | 29.5 | 44.0 | 43.1 | 33.3 |
| Pavllo [36] (‡) | **17.5** | 12.3 | 37.4 | **27.7** | 19.0 | 19.0 | 27.7 | 43.4 | 42.5 | 33.0 |
| Ours | 18.9 | **12.3** | 38.1 | 28.5 | **18.1** | **18.2** | **27.1** | **40.9** | **40.2** | **32.2** |

Table A1: Results on HumanEva-I for multi-action (MA) models reported in Protocol 1 (MPJPE), lower the better. ‡ indicates test time augmentation.

*Equivariance for $\gamma$, and $\beta$* is obtained by following the principle applied to fully connected layers: we achieve equivariance via parameter sharing and odd symmetry:

$$\gamma = \begin{bmatrix} [\gamma_{\mathtt{ln}} & \gamma_{\mathtt{lp}}] & [\gamma_{\mathtt{ln}} & \gamma_{\mathtt{lp}}] & [\gamma_{\mathtt{cn}} & \gamma_{\mathtt{cp}}] \end{bmatrix}^T \text{ and } \beta = \begin{bmatrix} [\beta_{\mathtt{ln}} & \beta_{\mathtt{lp}}] & [-\beta_{\mathtt{ln}} & \beta_{\mathtt{lp}}] & [\mathbf{0} & \beta_{\mathtt{cp}}] \end{bmatrix}^T.$$

*Equivariance for $\mu$, and $\sigma$* is obtained by computing the mean and standard deviation on the "augmented batch" and by keeping track of its running average. Formally, given a batch $\mathcal{B}$ of data,

$$\mu = \frac{1}{2|\mathcal{B}|} \sum_{\mathbf{x} \in \mathcal{B}} \mathbf{x} + \mathcal{T}^{\mathtt{in}}(\mathbf{x}), \quad \sigma = \sqrt{\frac{\sum_{\mathbf{x} \in \mathcal{B}} (\mathbf{x}-\mu)^2 + (\mathcal{T}^{\mathtt{in}}(\mathbf{x})-\mu)^2}{2|\mathcal{B}|}}.$$

### B.5 Dropout.

At test time, dropout scales the input by $p$, where $p$ is the dropout probability. The equivariance property is satisfied because of the associativity property of a scalar multiplication. The input and output dimension and symmetry of a dropout layer are identical. Therefore, $\mathcal{T}^{\mathtt{out}}$ and $\mathcal{T}^{\mathtt{in}}$ are identical. From the definition:

$$\mathcal{T}^{\mathtt{out}}(p \cdot \mathbf{x}) = \mathcal{T}^{\mathtt{in}}(p \cdot \mathbf{x}) = T^{\mathtt{in}}_{\mathtt{neg}} T^{\mathtt{in}}_{\mathtt{swi}}(p \cdot \mathbf{x}) = p \cdot (T^{\mathtt{in}}_{\mathtt{neg}} T^{\mathtt{in}}_{\mathtt{swi}} \mathbf{x}) = p \cdot (\mathcal{T}^{\mathtt{in}}(\mathbf{x})) \ \ \forall \mathbf{x} \in \mathbb{R}^{|J^{\mathtt{in}}||D^{\mathtt{in}}|}.$$

Hence, a dropout layer naturally satisfies the equivariance property. At training-time, we do not enforce equivariance for the dropped units, *i.e.*, we do not jointly drop symmetric units as we found this to prevent overfitting. This is likely application dependent.

## C  Additional Results

### C.1  3D pose estimation

In Tab. A1, we report the HumanEva-I for multi-action models evaluated on Protocol 1 (MPJPE). Our approach have benefits the most from the Boxing action while maintaing the performance on other actions. We also provide qualitative evaluation in Fig. A1 and Fig. A2. We observe that our model successfully estimates 3D poses from 2D key-points. We have also attached animations in the supplemental.

### C.2  Skeleton based action recognition

In Fig. A3, we show the visualization of the input skeleton sequences computed by OpenPose [2] and the predicted action class by our chiral invariant skeleton based action recognition model.

## D  Implementation Details

### D.1  3D pose estimation

**Implementation details.** Our model follows the temporal convolutional architecture proposed by Pavllo et al. [36], and replaced all layers with their chiral versions; code for the layers are attached in the supplemental as well. We also changed ReLU to tanh to achieve chiral equivariance. For the temporal models, we follow their 4 blocks design which has the receptive field of 243. For the single frame model, we follow their 3 blocks design. These models all contains 1020 hidden dimensions so it is a factor the number of joints, 17, this is slightly smaller than the 1024 used in [36]. We also use their data processing and batching stragety as described in Section 5 and Appendix A.5 of [36]. For

Figure A1: Qualitative visualization of 2D to 3D pose estimation for the action "Walking" on HumanEva-I dataset.

training the model, we utilized the Adam optimizer with beta1=0.9 and beta2=0.9999. We decay the batch-normalizations' momentum as suggested in [36]. Other details follows the publicly available implementation by Pavllo et al. [36]. We enforced chiral equivariance by choosing the $|D_n^{\text{out}}|$ to be $\frac{1}{3}$ of the hidden dimension. The $|D_n^{\text{in}}|$ for the input layer is 17 and the $|D_n|^{\text{out}}$ for the output layer is 17, as one for each joint.

## D.2   2D pose forecasting

**Implementation details.** The non-chiral equivariant baseline is a seq2seq model consisting of an encoder and decoder, which are stacked-LSTMs with hidden size of 1040 and 2 stacked layers. We trained using teacher forcing with the Adam optimizer. The batch-size is 256, and we trained for 30 epochs. Dropout is applied to the LSTMs' hidden layer with drop probability of 0.5. Following prior works, we use max norm gradient clipping of 5, a learning rate of 0.005 with a decay of 0.95 every 2 epochs. The data processing and evaluation setting follows [5]. Other details follows the publicly available implementation by Chiu et al. [5]. We enforced chiral equivariance by choosing the $|D_n^{\text{out}}|$ to be $\frac{1}{2}$ of the hidden dimension, as the output is two dimensional per joint.

Figure A2: Qualitative visualization of 2D to 3D pose estimation for the action "Boxing" on HumanEva-I dataset.

### D.3 Skeleton-based action recognition

**Implementation details.** The non-chiral version of the model, Ours-Conv, follows Temporal-Conv [21] while we modified the model to have not only temporal convolution but also spatial convolution. There are ten spatial-temporal convolution blocks and each block we first perform spatial convolution and then temporal convolution. The temporal convolution considers the intra-frame information while the spatial convolution considers the inter-frame information. For the recognition task, we need chiral invariance, *i.e.*, a chiral pair should be classified as the same action class. To this end, we use a chiral invariance layer where we let both $J_r^{\text{out}}$, $J_l^{\text{out}}$ as well as $D_n^{\text{out}}$ to be empty sets, which means there are no left and right joints but only center joints and there is no dimension that will be negated in the output of the layer after applying the chirality transform. Note that the chiral transformation exchange the left and right joints and negate the dimension in the index set $D_n^{\text{out}}$. Given $J_r^{\text{out}}$, $J_l^{\text{out}}$ and $D_n^{\text{out}}$ are all empty, it's obvious that the output will be chiral invariance. For the chiral invariance model, Ours-Conv-Chiral, we replace the all the non-symmetric layers before the chiral invariance layer with their corresponding chiral equivariance version. All the layers after the chiral invariance layer remains the same as in the Ours-Conv model. Similar to [21], there are in total 10 convolution blocks in Ours-Conv and we put the chiral invariance layer at the fourth layer. Also, we gradually reduce the ratio of the dimension to be negated ($|D_n^{\text{out}}|/|D^{\text{out}}|$) from

Push up

Clean and jerk

Juggling balls

Playing piano

Jogging

Figure A3: Visualization of the input skeleton sequences and the corresponding predicted action classes of our method on the Kinetics-400 dataset [20].

$\frac{1}{3}$ to $\frac{1}{6}$ at the first layer, from $\frac{1}{6}$ to $\frac{1}{12}$ at the second layer and from $\frac{1}{12}$ to $0$ at the third layer. We use the SGD optimizer with a momentum of $0.9$ as in [51] with a batch size of 256. We train the model for 90 epochs.



[Supplementary Material 2]

# Supplemental Materials

## Overview

This codebase contains implementations for "Chirality Nets for Human Pose Regression".

- Chiral layers are implemented under `pose_chiral/chiral_layers`.
- A tutorial on chirality equivaraince is under `demo/`.
- Test cases are implemented under `tests/`.

## Dependencies

- Python 3+
- Pytorch >= 1.0
- unittest

## How to run tests

Run the following in the `supp_code` directory. ˷ nosetests –nocapture ˷

## Expected results.

```
Tests batchnorm equivariance at test time.
Difference expected chiral pairs: 0.0

.Tests batchnorm running mean and var updates.
.Test equivariance for conv1d layer, different in/out sym_group.
Difference expected chiral pairs: 1.1701751e-13

.Test equivariance for conv1d layer, sym_group [1,1,1].
Difference expected chiral pairs: 6.711298e-14

.Test equivariance for conv1d layer, sym_group [2,2,1].
Difference expected chiral pairs: 2.1044277e-13

.Tests equivariance on linear layer.
Difference expected chiral pairs: 0.0
```

```
.Tests equivariance of GRU.
Difference expected chiral pairs: 2.6402491e-15

.Tests equivariance of LSTM.
Difference expected chiral pairs: 3.608225e-15

.
----------------------------------------------------------------------
Ran 8 tests in 0.225s

OK
```