[Reviews · NeurIPS 2019]

Reviewer 1



This paper presents the novel Chirality Nets where pose symmetry (chirality equivariance) is directly built into the networks. The proposed method has fewer trainable parameters and lower computational complexity. Extensive experiments on three different tasks show the effectiveness of the proposed method. The idea of parameter sharing is not novel, however, this paper designs a series of novel variants of the standard building blocks. The idea of built-in chirality equivariance is well motivated and interesting. Chirality equivariance for human pose regression is of great importance and interest to the community. Extensive experiments on various tasks show the wide range of potential applications of the proposed method. The paper is reasonably well-written and easy to follow. The authors also provide codes to ensure the reproducibility. Questions: (1) Will parameter sharing cause a loss of model representation power? Table 1, 2 and 3, it seems that the proposed Chirality Nets will (slightly) outperform the test-time augmentation baseline. Why? Can parameter sharing be viewed as a kind of model regularization to reduce overfitting? (2) The reviewer is also curious about the 2d human pose estimation performance of the proposed method. In the 2d human pose estimation tasks, there exist large in-the-wild datasets (MSCOCO and MPII dataset), where the overfitting problem is not significant. Will Chirality Nets achieve good results in such cases? =================UPDATE==================== After reading the comments and the author responses, most of my concerns are addressed. Overall this is a good paper. I will raise my rating from 6 to 7.

Reviewer 2



A growing body of literature has shown that building symmetries into neural networks through equivariant layers is an effective means of improving results, especially in the face of limited data and even when data augmentation is used. This paper continues that trend by showing that equivariance to chirality transformations consistently improves results on pose regression tasks. The paper is well written and easy to follow. Related work is discussed in a mostly adequate and balanced manner. The work fits in existing theoretical frameworks when considering that the group acts in a linear way (though not via permutations). Such networks are covered by the theory of Shawe-Taylor and colleagues (e.g. "Representation theory and invariant neural networks") as well as recent work by Kondor, Cohen, and others. This paper however focuses on the practical aspects of implementing chirality-equivariant layers, rather than mathematical theory, and as such makes a very useful contribution. It is shown that the equivariant layers reduce the number of parameters and FLOPS. A very solid experimental validation is performed, showing consistent improvements over recent state of the art methods for this task. The improvements are not very large, but this is not to be expected from such a small (2 element) symmetry group. Overall, this is a nice paper with a simple, well executed idea. Typo on line 75: equvariant >>>> Post rebuttal comments I have read the other reviews and the rebuttal. The reviewers seem to agree that this paper makes a useful contribution and should be accepted. Since I did not raise any major concerns in my initial review, the rebuttal was mainly addressed at the other reviewers, and so did not change my judgement significantly.

Reviewer 3



This paper can be considered as the first to apply chirality into the network structure design and has proved its effectiveness in some tasks that are related to chirality transform. It may inspire a set of work utilizing this property in the field, which potentially has large impact. The empirical experiments mostly indicates the effectiveness of the new network structure. It would be better if it can show the performance for the entire human pose estimation pipeline, i.e., 2D pose estimation and 2D to 3D mapping. It can be achieved by modifying the state-of-the-art network, i.e. hourly-glass, into its chiral form. Besides, the network runtime and memory consumption can be revealed by quantitative results. It would be better if the experiments could be more complete. Overall, I think this is a good paper with strong potential to benefit the readers. It has well-organized structure and good clarification, which worths a clear accept.

[Author Response · NeurIPS 2019]

**General response:** We thank all reviewers for their comments. The reviewers agree that the idea of the proposed chiral equivariance is well motivated and interesting (R1) and the paper can be considered as the first to have a chirality transform built into the network structure (R3). The paper focuses on the practical aspects rather than mathematical theory, which makes a very useful contribution (R2) and may inspire a set of work utilizing this property in the field, which potentially has large impact (R3). A very solid/extensive experimental validation is performed (R1&R2&R3). Moreover, the paper is well written, easy to follow (R1&R2) and well-organized (R3). The authors also provide code to ensure reproducibility (R1). In the following we address all comments individually.

**Reviewer # 1:**

**Q1:** *Novelty of the technique to achieve equivariance.* Note that chirality equivariance cannot be achieved with prior methods that use parameter sharing to obtain equivariance, *e.g.*, [38]. Beyond parameter sharing, we introduce *odd and even* symmetry: note for example the negative signs in $W$ for fully connected layers discussed in L145.

**Q2:** *On the loss of model representation power.* Empirically, on chiral data, we do not observe a loss of representation power for chirality nets, even though there are less trainable parameters. Throughout all our experiments, the baselines and their chiral counterparts have the same number of layers. Hence, the chiral net has less trainable parameters for each layer. On three different tasks (2D to 3D pose estimation, 2D pose forecasting and skeleton based action recognition), we observe the chiral net to outperform the baselines despite fewer trainable parameters. Obviously, if a chiral net were applied to tasks that don't exhibit chirality, we expect the representation power to be insufficient.

**Q3:** *Can parameter sharing be viewed as a kind of model regularization to reduce overfitting?* Yes, this is a compelling interpretation. It can also be viewed as reducing the hypothesis space of the model which decreases sample complexity according to statistical learning theory insights.

**Q4:** *2d human pose estimation performance of the proposed method.* Chirality equivariance is valid for skeletons like human poses, *i.e.*, 2D or 3D key-points. The input for the task of 2D human pose estimation is an image (represented on a 2D-grid). Chirality is not a suitable property for this input data. However, it is certainly an interesting question how to extend the chirality equivariance definition to handle image data.

**Reviewer # 2:** We thank Reviewer # 2 for appreciating the paper. We'll fix typos.

**Reviewer # 3:**

**Q5:** *Jump from sec 3.2 chirality transformation directly into the fully connected layer not smooth. More clarification for better understanding the weight design.* We'll add a description at the beginning of Sec. 3.3, explaining why we need to share the parameters in the form specified in L145. Intuitively, in order to achieve chirality equivariance, we treat the left joints and the right joints equally. Therefore the parameters are shared for joints in the left and right group. Next, to handle the reflection in pose, certain dimensions are negated. We utilized odd symmetry in the parameters to "cancel" the negation. Note, we also provided the proof that verifies the chiral equivariance of the proposed layer in the supplementary material **B.1**. We are happy to extend.

**Q6:** *Why not verify the proposed network on the 2D pose estimation task?* Please see **Q4.**

**Q7:** *How does the 3D pose accuracy influence the action recognition accuracy.* We think the accuracy of 3D pose estimation is one of the important factors that influences the performance of skeleton-based action recognition. However, the focus of the paper is to discuss and demonstrate the effectiveness of the proposed chiral nets. Therefore, we used the same input 3D pose and evaluation procedure as prior works. We leave the study of robustness to noise in 3D pose estimation to future work.

**Q8:** *For limited data setting, the paper lacks an in-depth reasoning why when the training data ratio is extremely small, the performance is inferior to [36], while further increase training data the performance surpasses [36].* We studied this question and report our thoughts in L238-L240. We think that the hyper-parameters are not optimal for this extreme case, in particular, a batch-size of 64, used for the large datasets. As also discussed in [1], a smaller batch-size generally leads to better generalization error. We think that, the "small"-ness of a batch is relative to the size of the dataset. To verify this, we further reduced the batch-size to 16. We observe that Pavllo *et al.* [36] achieves 103.8mm and our approach improved to 98mm. We think that due to parameter sharing, the "effective batch-size" is twice that of the baseline. Note: for a fair and consistent comparison, we used the same hyper-parameters (batch-size of 64) following the setting in [36] for all models in the limited data experiments.

**Q9:** *Quantitative experiments on the memory saving and computation saving.* Empirically, the running time is highly dependent on the implementation and hardware platform; which deserves a careful study that is beyond the scope of this work. For this reason, we analyze the time complexity and model complexity in Sec. 3.4 and show that chirality nets are more efficient as there are less FLOPs and less model parameters due to symmetry. We analyze the model complexity of the chiral net in the paper L174-L176 and we discuss the FLOPs of a chiral net in the paper L177-L184: we show that the number of parameters is reduced by a factor of $\frac{|(|J_1^{\text{in}}|+|J_c^{\text{in}}|)\cdot(|J_1^{\text{out}}|+|J_c^{\text{out}}|)}{|J^{\text{in}}|\cdot|J^{\text{out}}|}$ in each chiral layer and the number of multiplications reduces by a factor of $\frac{|J_1^{\text{in}}|+|J_c^{\text{in}}|}{|J^{\text{in}}|}$.

**References:**
[1] N. S. Keskar, D. Mudigere, J. Nocedal, M. Smelyanskiy, and P. T. P. Tang. On large-batch training for deep learning: Generalization gap and sharp minima. In *Proc. ICLR*, 2017.


[Meta-Review · NeurIPS 2019]

The paper proposes Chirality Nets for human pose regression, where the resulting network layers are equivariant to chirality transformations. Overall the paper presents some significantly novel results in a well written and intuitive manner. The experimental evaluation convincingly supports the proposed approach. The reviewers and AC consistently agree that the submission is of significant interest and novelty, and that the authors feedback has adequately addressed the points raised in the reviews.